# Direct Diagnosis of Echovirus 12 Meningitis Using Metagenomic Next Generation Sequencing

**DOI:** 10.3390/pathogens10050610

**Published:** 2021-05-17

**Authors:** Madjid Morsli, Christine Zandotti, Aurelie Morand, Philippe Colson, Michel Drancourt

**Affiliations:** 1IHU Méditerranée Infection, 13005 Marseille, France; mor_madjid@hotmail.com (M.M.); christine.zandotti@ap-hm.fr (C.Z.); philippe.colson@univ-amu.fr (P.C.); 2Institut de Recherche pour le Développement (IRD), Microbes Evolution Phylogeny and Infections (MEPHI), Aix-Marseille-Université, IHU Méditerranée Infection, 13005 Marseille, France; 3Service de Pédiatrie, Hôpital de la Timone, Assistance Publique à Marseille, 13005 Marseille, France; aurelie.morand@ap-hm.fr

**Keywords:** enterovirus meningitis, whole genome sequencing, metagenomic next-generation sequencing, *Echovirus* 12, cerebrospinal fluid

## Abstract

The current point-of-care diagnosis of enterovirus meningitis does not identify the viral genotype, which is prognostic. In this case report, more than 81% of an *Echovirus* 12 genome were detected and identified by metagenomic next-generation sequencing, directly from the cerebrospinal fluid collected in a 6-month-old child with meningeal syndrome and meningitis: introducing *Echovirus* 12 as an etiological agent of acute meningitis in the pediatric population.

## 1. Introduction

The highly diverse viral genus *Enterovirus* encompasses more than 300 genotypes [1]. Some *Enterovirus* members are responsible for central nervous system (CNS) infections, which clinical and epidemiological characteristics and prognosis, vary according to the precisely identified enterovirus, and some genotypes have been associated with a particular clinical severity and mortality [2]. The *Enterovirus* genotype is not routinely determined by molecular diagnosis assays detecting enterovirus RNA at the core and point-of-care (POC) laboratories [3,4,5,6]. *Echovirus* strains belonging to *Enterovirus B* species preferentially infect infants and young children [1,4] and are frequently involved in aseptic meningitis and encephalitis. Indeed, *Enterovirus* genotyping is most commonly based on partial VP1 gene Sanger sequencing using a generic standard protocol which is not commonly applied during the time of care [7].

In order to challenge *Enterovirus* genotyping at the POC laboratory, we herein developed a unique protocol to diagnose and genotype *Enterovirus* CNS infection directly from the cerebrospinal fluid (CSF) using metagenomic Next-Generation Sequencing (mNGS). This diagnosis approach is here illustrated by the diagnosis of *Echovirus* 12 meningitis in a child, a rarely reported situation in such setting [8,9].

## 2. Case Presentation and Methods

A 6-month-old girl born from twin pregnancy was admitted at the emergency department with fever (37.7 °C), cough and meningeal syndrome. She had an history of Respiratory Syncytial Virus bronchiolitis that led to hospitalization one month prior to meningitis. At the admission, CSF analysis after lumbar puncture showed a leukocyte count of 1 cell/mm^3^, protein at 0.14 g/L and glucose at 3.41 mmol/L. Microscopic analysis after Gram staining was negative. Investigation of the CSF at the POC laboratory using the Biofire FilmArray Meningitis/Encephalitis panel (bioMérieux, Marcy-l, Etoile, France) [6] was positive for *Enterovirus*.

In parallel, total RNA was manually extracted from 200 µL of CSF, following an in-house developed protocol, using QIAamp Viral RNA Mini Kit solutions (Qiagen, Hilden, Germany) for lysing, and washing steps, then total RNA was then purified using RNA specific magnetic Dynabeads (Life technology, Oslo, Norway). Briefly, 200 µL of CSF were incubated with 50 µL proteinase K (Qiagen) for 5 min at room temperature, then 300 µL AVL lysis buffer (Qiagen) were added and incubated for 15 min at room temperature. A 150-µL volume absolute ethanol (99%) were added to the lysis mix, 50-µL Dynabeads (40 mg/µL) were added and incubated for 15 min at room temperature. The Dynadeads/RNA complex washed twice with 850 µL AV1 solution (Qiagen), then two times with 450 µL AV2 solution (Qiagen) in the presence of 70% ethanol. After the second wash, Dynabeads were dried for 15 min at room temperature and eluted in a 60 µL-volume, then incubated for 3 min at 70 °C, followed by magnetic separation. Finally, 3 µL of RNaseOUT™ Recombinant Ribonuclease Inhibitor (Invitrogen, Illkirch, France were added to the purified RNA was and stored at −70 °C.

A 40-µL volume of total RNA was treated with ezDNase (Invitrogen, Illkirch, France) and concentrated with (Zymo Research, Irvine, CA, USA) kit, then eluted in 20 µL sterile water (Figure 1). The complementary DNA (cDNA) synthesis was performed using TaqMan kit according to the manufacturer protocol (Applied Biosystem, Foster City, CA, USA) in 50µL containing 19.25 µL eluted RNA, then 20 µL of the cDNA were used as a matrix for double strand synthesis, using 3 units of DNA Polymerase I, Large (Klenow) Fragment (BioLabs) in a 30 µL-volume. Double stranded DNA purified with Agencourt^®^ AMPure beads (Invitrogen) and eluted in 17 µL of 1x-sterile Tris-EDTA solution. Finally, 1 ng of cDNA was used for metagenomics Next-Generation Sequencing (mNGS) library preparation (Appendix A), using Illumina Nextera XT paired-end protocol (Illumina, San Diego, CA, USA), as previously described [10,11] and sequenced on iSeq 100 instrument in a single 17.5-h run providing 2 × 150-base pair (bp) long reads. The NGS generated sequences were assembled by Spades on-line software available on Galaxy/Europe bioinformatics (https://usegalaxy.eu, accessed on 17 June 2020) and mapped with CLC Genomics Workbench software version 7.5.0 (Qiagen).

## 3. Results and Discussion

BLAST analysis of the contigs generated by mNGS after assembling reads with Spades (https://usegalaxy.eu, accessed on 17 June 2020), yielded as best match the *Echovirus 12* strain Travis 2–85 gene (GenBank accession number AF295499.1). This strain was originally isolated from a 6-year-old healthy American male, caused cytopathic effect in tissue culture, was not neutralized by poliomyelitis antiserum, and failed inducing disease in infant mice [12]. As the complete genome of this strain was not available in the GenBank database, the *Echovirus 12* complete genome, prototype Travis (X77708.1) was used as reference sequence for mapping of total reads by CLC Genomics Workbench software. The iSeq sequencing generated 114.818 reads, and 76.284 (66.4%) reads could be mapped on the *Echovirus 12* genome, generating 7 contigs (GenBank accession number; PRJEB39568) covering 6,127 bp, hence 81.7% of this genome (Figure 1). Phylogenetic analysis based on VP1 gene and 3D polymerase encoding genes identified that these sequences belong to an *Echovirus 12* as supported by bootstrap values of 98% and 97%, respectively (Figure 2, Appendix A).

Using this approach of whole genome sequencing, *Echovirus 12* was obtained in one-shot protocol directly from CSF sample. To confirm the NGS result, two *Echovirus 12*-specific primers were designated and used to target a 291-bp *Echovirus 12* genome fragment. The same strain of *Echovirus 12* was identified by sequencing of the amplified fragment at 98.44% sequence identity (not published data).

*Echovirus 12* has been detected in patients with diarrhea and aseptic meningitis [9,13]. In this study, we described for the first time a clinical case of *Echovirus 12* meningitis diagnosed by near whole genome sequencing of an *Echovirus 12* directly from CSF by mNGS. This strategy proves high sensitivity in enterovirus detection, which warrants its introduction for routine diagnostic of enterovirus meningitis in addition to viral genomic surveillance and may even be considered for POC laboratories if the very fast Oxford Nanopore Technology is used. Current routine diagnostic targets a short sequence covering around 7% of the *Enterovirus* genome, so that is not indicative for *Enterovirus* genotyping [11]. The availability of the genome instead of a gene fragment necessarily provides improved information regarding typing of the viral strain and identifying mutations and recombinations, and correlating these genotypic features with epidemiological and/or clinical ones. Another important benefit of mNGS upon qPCR or Sanger sequencing is its versatility. Indeed, this is not a targeted approach but instead it is an opened approach that can potentially detect sequences from any virus or microorganism provided these are in sufficient amount; this is thus of particular interest in cases when no infectious agent could be diagnosed. In addition, NGS cost per clinical sample is currently in the same order of magnitude than that of Sanger sequencing. Indeed, we have estimated that current cost per nucleotide is EUR 0.3 for Sanger sequencing and EUR 0.03 for NGS sequencing with Illumina technology; such cost cannot be directly extrapolated to other laboratories due to highly variable cost components, among which the commercial policy of suppliers or the infrastructure of the laboratory. Overall, the present case exemplifies the powerfulness of mNGS in the setting of the routine diagnosis of meningitis.

## Figures and Tables

**Figure 1 pathogens-10-00610-f001:**
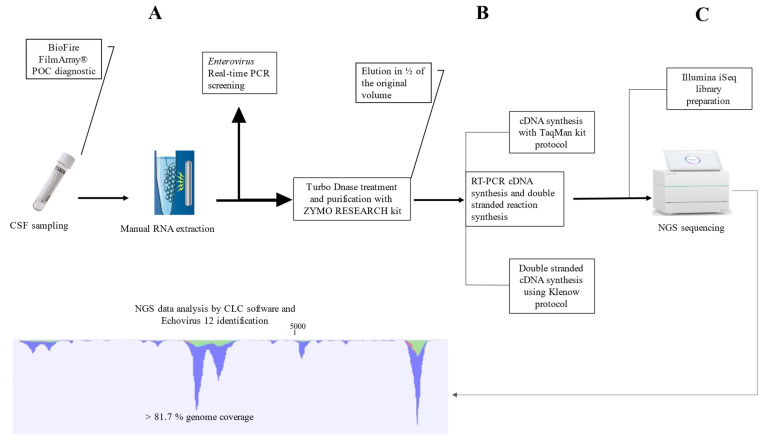
*Echovirus* 12 whole genome sequencing direct diagnosis workflow. The total duration of the handling process is less than 24 h including al steps. (**A**) CSF sampling by lumbar puncture, followed by specific point-of-care (POC) enterovirus (EVs) diagnosis in FilmArray^®^ CSF direct test (BioFire Diagnostics, Salt Lake City, UT, USA). The viral RNA was extracted and purified following an in-house developed protocol, using QIAamp Viral RNA Mini Kit (Qiagen) and RNA was purified by magnetic DynaBeads (Thermo Fisher, Waltham, MA, USA). The EVs POC diagnostic was confirmed by real-time PCR using LightCycler Multiplex RNA Virus Master kit (Roche Diagnostics^®^, Mannheim, Germany). (**B**) The extracted RNA was treated by Turbo DNase (Thermo Fisher) and purified with (Zymo Research) kit. RT-cDNA synthesis reaction was performed in 50 µL-volume using kit TaqMan (Thermo Fisher), followed by double strand synthesis using DNA Polymerase I, Large (Klenow) Fragment (BioLabs). (**C**) The double stranded cDNA sequenced in 150-cycle iSeq Illumina instrument following Nextera NGS library preparation iSeq protocol. Finally, NGS data analysis was performed using CLC Genomics Workbench software version 7.5.0 (Qiagen), and more than 81.68% of *Echovirus* 12 genome was obtained directly by next-generation sequencing.

**Figure 2 pathogens-10-00610-f002:**
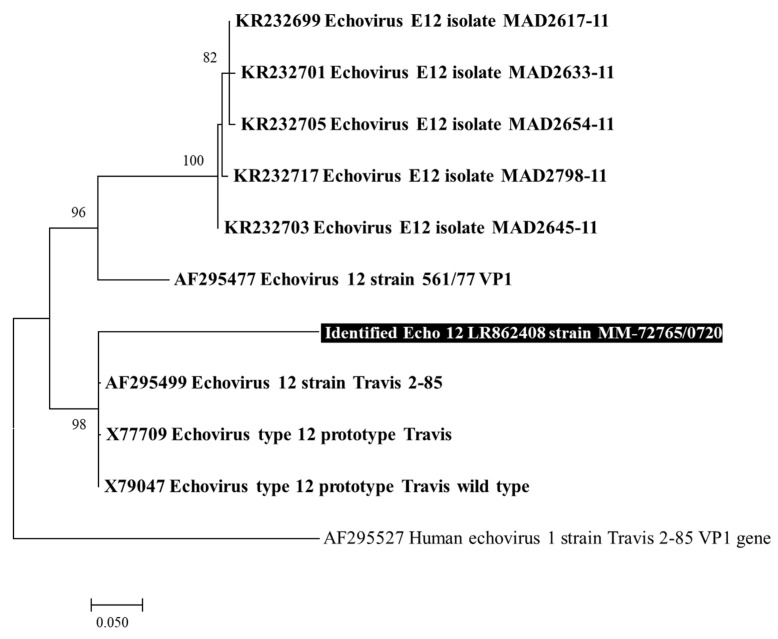
Phylogenetic analysis based on *VP1* gene. The *Echovirus 12* near full-length genome obtained here by mNGS (GenBank accession no LR862408; indicated by a white bold font and a black background, and 9 hit blast *Echovirus 12* recovered from the NBCI GenBank nucleotide sequence database (http://www.ncbi.nlm.nih.gov/nucleotide/, accessed on 29 April 2021), indicated by a bold font, were incorporated in the phylogeny reconstruction in addition to *Echovirus 1* sequence recovered from GenBank database. The sequence obtained in the present study is most similar to *Echovirus 12* Travis stains (X79047, X77709, AF295499) and are clustered with this sequence, confirming the BLAST result. The evolutionary history was inferred in the MEGA 7 software version 7.0.2. This analysis involved 11 nucleotide sequences. There were a total of 548 positions in the final dataset. The tree was performed by applying the neighbor-joining method and the Kimura 2-parameter method. The percentage of replicate trees in which the associated taxa clustered together in the bootstrap test (1000 replicates) is shown next to the branches. The tree was drawn to scale, with branch lengths in the same units as the evolutionary distances used to infer the phylogenetic tree; the scale bars represent the corresponding number of nucleotide substitutions per site. Bootstrap values ≥ 75% are indicated at the nodes.

## Data Availability

The identified *Echovirus 12* genome available on NCBI GenBank accession number; PRJEB39568.

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
