# Peer review of "Direct Diagnosis of Echovirus 12 Meningitis Using Metagenomic Next Generation Sequencing"

_pathogens, 2021, doi:10.3390/pathogens10050610_

Round 1
Reviewer 1 Report
The authors utilize metagenomics sequencing to inspect the genome of an echovirus 12 infection in a six-month old patient afflicted with meningitis. Echoviruses are one of many strains in the Enterovirus B species which can cause minor illnesses or, more rarely, infect the central nervous system and cause serious diseases, including meningitis, paralysis, and other neurological complications. Echoviruses preferentially infect in infants and young children. The investigators described the diagnosis and isolation procedure of echovirus 12 from the cerebral spinal fluid, RNA extraction and assembling of NGS generated sequences to compile a near complete sequence of the patient viral genome. From the study, new information regarding the genomic information of an echovirus rarely associated with CNS infection can now be analyzed. The authors performed phylogenetic analysis of the viral genome and show close sequence similarity to Echovirus 12 (Travis), a strain originally isolated from patients causing cytopathicity in tissue culture, not neutralized by poliomyelitis antiserum, and failing to induce disease in infant mice (Hammon et al, 1957). The short study sheds light on Echovirus 12 association with meningitis in infants, and the recognition that more research is necessary to determine how these relatively common viruses may cause neurological complications. The viral genomic data submitted to GenBank will assist future bio-informatic investigators in identifying changes in viral genes responsible for pathogenesis.
The authors should include relevant information (as described above) on the echovirus 12 Travis strain, and implications with their GenBank deposited strain.
Author Response
Reviewer 1.
1/ The authors utilize metagenomics sequencing to inspect the genome of an echovirus 12 infection in a six-month old patient afflicted with meningitis. Echoviruses are one of many strains in the Enterovirus B species which can cause minor illnesses or, more rarely, infect the central nervous system and cause serious diseases, including meningitis, paralysis, and other neurological complications. Echoviruses preferentially infect in infants and young children. The investigators described the diagnosis and isolation procedure of echovirus 12 from the cerebral spinal fluid, RNA extraction and assembling of NGS generated sequences to compile a near complete sequence of the patient viral genome. From the study, new information regarding the genomic information of an echovirus rarely associated with CNS infection can now be analyzed. The authors performed phylogenetic analysis of the viral genome and show close sequence similarity to Echovirus 12 (Travis), a strain originally isolated from patients causing cytopathicity in tissue culture, not neutralized by poliomyelitis antiserum, and failing to induce disease in infant mice (Hammon et al, 1957). The short study sheds light on Echovirus 12 association with meningitis in infants, and the recognition that more research is necessary to determine how these relatively common viruses may cause neurological complications. The viral genomic data submitted to GenBank will assist future bio-informatic investigators in identifying changes in viral genes responsible for pathogenesis.
Authors’ answer: We are thankful to the reviewer for these comments. This manuscript was amended in accordance with the recommendations and comments of the reviewers. We provided our item by item responses to each point brought up, inserted underneath each of them. Parts that are amended in the revised manuscript are indicated with a red font in a manuscript file with marked revisions.
2/ The authors should include relevant information (as described above) on the echovirus 12 Travis strain, and implications with their GenBank deposited strain.
Authors’ answer: As advised by the reviewer, this information has been added in the revised manuscript, see on lines 94, 143-144.
Reviewer 2 Report
Specific comments:
Title can be improved. Maybe like: Diagnosis of Echovirus 12 meningitis using NGS ...or something similar...
Near whole genome is a not good way to put it.
Introduction ja discussion
I realize that there is a word limit but is it possible to elaborate the costs and benefits of NGS in routine diagnostic laboratories? Of would this NGS approach suitable for epidemiological purposes rather than for diagnosis? As the result "enterovirus" is good enough for clinic in many cases, and only in special cases (new manifestation etc) this NGS is good tool? Or what do you think about this?
Discussion
Why this NGS approach should be considered as part of future diagnostic laboratories? It costs and cannot be used routinely as it takes time to get the results. How these NGS pipelines should be validated and quality checked in routine laboratories?
I feel that point-of-care is more like diagnosis carried out next to the patient. So please do not use "point-of-care" in this NGS context. It is far from it. Check the manuscript and rephrase all of these.
Sentence: Echovirus 12 infection is rarely associated to CNS infection so that its prognosis is poorly documented (9). How this reference Rosenwirth and Eggers, 1978 (9) is linked to this sentence? Check better references. Is there any CNS and echo 12 studies/cases published? Later you state that there is no reports published linking CNS and E-12. Rephrase sentence accordingly.
Echoviruses are common agents causing meningitis, but if we specifically consider E-12; is there any reports published? Epidemics? Anything? Please, tell a little bit more but be precise.
Author Contributions
Finding -> Funding?
Author Response
Reviewer 2.
1/ Specific comments:
Title can be improved. Maybe like: Diagnosis of Echovirus 12 meningitis using NGS ...or something similar...
Near whole genome is a not good way to put it.
Authors’ answer: As advised by the reviewer the title has been changed for: “Direct diagnosis of Echovirus 12 meningitis using metagenomic next generation sequencing”" (line 2).
2/ Introduction ja discussion
I realize that there is a word limit but is it possible to elaborate the costs and benefits of NGS in routine diagnostic laboratories? Of would this NGS approach suitable for epidemiological purposes rather than for diagnosis? As the result "enterovirus" is good enough for clinic in many cases, and only in special cases (new manifestation etc) this NGS is good tool? Or what do you think about this?
Authors’ answer: As advised by the reviewer we have elaborated the costs and benefits of NGS in routine clinical microbiological and virological diagnosis laboratories. See lines 112-122: “The availability of the genome instead of a gene fragment necessarily provide improved information regarding typing of the viral strain and identifying mutations and recombinations, and correlating these genotypic features with epidemiological and/or clinical ones. Another important benefit of metagenomic NGS upon qPCR or Sanger sequencing is its versatility. Indeed, this is not a targeted approach but instead it is an opened approach that can potentially detect sequences from any virus or microorganism provided these are in sufficient amount; this is thus of particular interest in cases when no infectious agent could be diagnosed. In addition, NGS cost per clinical sample is currently in the same order of magnitude than that of Sanger sequencing. Overall, the present case exemplifies the powerfulness of metagenomic NGS in the setting of the routine diagnosis of meningitis.”.
3/ Discussion
Why this NGS approach should be considered as part of future diagnostic laboratories? It costs and cannot be used routinely as it takes time to get the results. How these NGS pipelines should be validated and quality checked in routine laboratories?
Authors’ answer: NGS has been recently introduced in clinical microbiology and virology laboratories, and adds to previous tools in routine diagnosis. As stated above and added in the discussion lines 112-122, “The availability of the genome instead of a gene fragment necessarily provide improved information regarding typing of the viral strain and identifying mutations and recombinations and correlating these genotypic features with epidemiological and/or clinical ones. Another important benefit of metagenomic NGS upon qPCR or Sanger sequencing is its versatility.”. “NGS cost per clinical sample is currently in the same order of magnitude than that of Sanger sequencing”. Moreover, its implementation at the clinical microbiology and virology laboratory for routine diagnosis has already started and is currently boosted by the SARS-CoV-2 and its use to diagnose SARS-CoV-2 variants. The validation and quality check of NGS tests in routine laboratories follow overall the same rules than those of other biological diagnostic tests.
4/ I feel that point-of-care is more like diagnosis carried out next to the patient. So please do not use "point-of-care" in this NGS context. It is far from it. Check the manuscript and rephrase all of these.
Authors’ answer: We have implemented point-of-care laboratories in our institution since 2008 and currently performed about 50,000 tests yearly for about thirty pathogens. We are therefore aware of POC testing. We removed POC from the sentence as we describe here the use of metagenomic NGS using Illumina technology. However, metagenomic NGS with the Nanopore technology can provide results very fast, within a few minutes in some cases as previously reported (Moon et al., Emerg Infect Dis 2018, doi: 10.3201/eid2410.180234). We clarified this point lines 107-111: “This strategy proves high sensitivity in enterovirus detection, which warrants its introduction for routine diagnostic of enterovirus meningitis in addition to viral genomic surveillance and may even be considered for POC laboratories if the very fast Oxford Nanopore Technology is used.”.
5/ Sentence: Echovirus 12 infection is rarely associated to CNS infection so that its prognosis is poorly documented (9). How this reference Rosenwirth and Eggers, 1978 (9) is linked to this sentence? Check better references. Is there any CNS and echo 12 studies/cases published? Later you state that there is no reports published linking CNS and E-12. Rephrase sentence accordingly.
Authors’ answer: To our best knowledge, no case of Echovirus 12 meningitis has been reported so far. In the reference cited, the authors indicated that it could be associated with meningitis, but this was not documented. The sentence has been corrected accordingly lines 103-107).
6/ Echoviruses are common agents causing meningitis, but if we specifically consider E-12; is there any reports published? Epidemics? Anything? Please, tell a little bit more but be precise.
Authors’ answer: According to the literature no case Echo-12 had been reported CNS infection, despite the echoviruses are frequently associated with meningitis “Track Changes" (line 103-107).
7/ Author Contributions
Finding -> Funding?
Authors’ answer: Corrected accordingly “Track Changes" (line 129).
Reviewer 3 Report
The manuscript by Madjid et al describes the characterization of an E-12 virus isolated from a case with meningitis using NGS. However the title does not reflect this and should be changed (eg simply adding meningitis case is enough).
Introduction:
Due to the molecular characterization of many viruses, the authors should refrain from using serotypes. Rather types or genotypes is the common word used nowadays as serotypes reflect serological characterization.
Remove “ among Enterovirus “ in the sentence: Some Enterovirus members are responsible for central nervous system (CNS) infections, which clinical and epidemiological characteristics and prognosis, vary according to the precisely identified virus among Enterovirus
Please also describe the POC real time assay as this is of value to the results where the authors indicate an E-9 with this method
The authors use the illumine platform which fragmentates the sequence. The authors also indicates identifying a E-9 by POC indicating a double infection and that the assembled virus might not be entirely E-12. The authors also only confirmed the E-12 virus and not the E-9 virus.
It is not clear on which region the phylogenetic tree is based? The shortest sequence is the partial travis genome which is 3.9 kb. Is that the region which was used?
The authors should do gene specific phylogenetic analysis as EVs, especially Echos can recombine which may obscure the true phylogenetic signal. I addition the authors included a clone of the travis strain which should never be included in clinical phylogenetic analysis as these are experimentally made and do not reflect E-12 circulation. There are enough clinical complete genome isolates as well as partial VP1 isolates to asses the phylogeny of the identified strains. txid35293[Organism:exp] - Nucleotide - NCBI (nih.gov) and txid35293[Organism:exp] complete - Nucleotide - NCBI (nih.gov).
Please include the tree as a main figure as well.
Has the sequence been submitted to genbank and the ENA database?
What was the sample to result time versus sanger sequencing?
I take it that this was not the only sample in the run? How do the authors guarantee no cross contamination, no bleeding of barcodes?.
Author Response
Reviewer 3.
1/ The manuscript by Madjid et al describes the characterization of an E-12 virus isolated from a case with meningitis using NGS. However, the title does not reflect this and should be changed (eg simply adding meningitis case is enough).
Authors’ answer: As advised by the reviewer the title has been changed for: “Direct diagnosis of Echovirus 12 meningitis using metagenomic next generation sequencing”" (line 2).
2/ Introduction:
Due to the molecular characterization of many viruses, the authors should refrain from using serotypes. Rather types or genotypes is the common word used nowadays as serotypes reflect serological characterization.
Authors’ answer: As relevantly suggested by the reviewer, the term “serotype” was deleted in the revised manuscript on line 1 of introduction and in the Figure legend.
3/ Remove “ among Enterovirus “ in the sentence: Some Enterovirus members are responsible for central nervous system (CNS) infections, which clinical and epidemiological characteristics and prognosis, vary according to the precisely identified virus among Enterovirus
Authors’ answer: Corrected accordingly “Track Changes" (line 21).
4/ Please also describe the POC real time assay as this is of value to the results where the authors indicate an E-9 with this method
Authors’ answer: The POC diagnosis was based on the BioFire FilmArray® assay according to the meningitis/ encephalitis panel (reference 6), which indicates the presence or absence of Enterovirus RNA in the CSF (Lines 39-41). The routine genotyping yielded Echo9 by Sanger small target sequencing (<300 bp) of VP1 gene.
The sentence on Echovirus 9 was amended for the purpose of clarification. As a matter of fact, although the fragment obtained by Sanger sequencing matched with an Echovirus 9, it was short (242 nucleotides) and its sequence was not found in the near full-length genome we obtained by NGS, which contains a counterpart sequence for this genome region that best matches with an Echovirus 12. In addition, we mapped the NGS reads generated from the CSF on an Echovirus 9 genome and only obtained a consensus sequence containing non-contiguous fragments with a total length of 1,950 nucleotides (instead of a near full-length genome when mapping on an Echovirus 12 genome) and this consensus sequence showed a better BLAST score when compared to our Echovirus 12 near full-length genome (score=3726) than to the most similar Echovirus 9 genome (LC321988.1; score= 1615).
5/ The authors use the illumina platform which fragmentates the sequence. The authors also indicate identifying a E-9 by POC indicating a double infection and that the assembled virus might not be entirely E-12. The authors also only confirmed the E-12 virus and not the E-9 virus.
Authors’ answer: The sentence on Echovirus 9 was amended for the purpose of clarification. As a matter of fact, the POC diagnostic assay (the BioFire FilmArray® assay according to the meningitis/ encephalitis panel) does not provide information on the Enterovirus genotype, just an Enterovirus-positive (or -negative) result. As explained in our response to the previous reviewer’s comment, we did not find Echovirus 9 sequence by NGS.
6/ It is not clear on which region the phylogenetic tree is based? The shortest sequence is the partial travis genome which is 3.9 kb. Is that the region which was used?
Authors’ answer: We clarified this point: the phylogenetic analysis involved 39 nucleotide sequences. There were a total of 4,533 positions in the final dataset.”. See line 95-97.
7/ The authors should do gene specific phylogenetic analysis as EVs, especially Echos can recombine which may obscure the true phylogenetic signal. I addition the authors included a clone of the travis strain which should never be included in clinical phylogenetic analysis as these are experimentally made and do not reflect E-12 circulation. There are enough clinical complete genome isolates as well as partial VP1 isolates to assess the phylogeny of the identified strains. txid35293[Organism:exp] - Nucleotide - NCBI (nih.gov) and txid35293[Organism:exp] complete - Nucleotide - NCBI (nih.gov).
Authors’ answer:
We are thankful to the reviewer for this suggestion. We performed VP1-based phylogeny accordingly. This new phylogenetic tree is now provided as appendix 1. The same cluster result was obtained by phylogenetic analysis based on the enterovirus VP1 gene (X nucleotides in length), classifying the VP1 region of the genome obtained in the present work as Echovirus 12, but bootstrap value was weaker (76% instead of 100%).
We used the Echovirus type 12, prototype Travis wild type genome GenBank: X79047.1 in the phylogenetic analyses.
8/ Please include the tree as a main figure as well.
Authors’ answer: The phylogeny analysis is added accordingly, “Track Changes" (line 95-97) and (figure 2).
9/ Has the sequence been submitted to genbank and the ENA database?
Authors’ answer: Yes the sequence has been submitted to GenBank: GenBank accession no LR862408. This has been mentioned in the manuscript in the Figure 1 legend.
10/ What was the sample to result time versus sanger sequencing?
Autors’ answer: We used here the iSeq illumina sequencer as a means of diagnosis and genotyping of enteroviruses, Its run lasts 17 hours. The overall procedure we used (from RNA extract to end of the NGS run took 24 hours including, 45 minutes of RNA extraction, 2 hours of RT+ klenow and 3 hours of NGS library preparation. Currently, we are implementing the mNGS in the POC laboratory using Oxford Nanopore technology on a MinION instrument, which can allow obtaining a diagnostic in less than 6 hours depending on the pathogen charge (Morsli et al, 2021).
11/ I take it that this was not the only sample in the run? How do the authors guarantee no cross contamination, no bleeding of barcodes?
Authors’ answer: for illumina library preparation, we limited the number of samples to 30 per run, then we separated each line of samples by an empty line, so the negative control from the beginning to the end of the library workflow.
For the barcoding we used the combinations recommended by the manufacturer, each sample has a different combination to avoid contamination
Round 2
Reviewer 3 Report
The manuscript has improved, but there are still a few issues that need to be adressed.
Please remove the figure based on the 4k tree. As indicated before due to reombination, complete genome trees for EVs is difficult. Instead please add the VP1 tree shown in the supp and add a 3Dpol tree. Remove all non E-12 strains and only include one other ECHO strain (eg E1 or E30) as outliers.
Line 118-120- This sentence in not correct. meningitis is a CNS related disease, yet the authors state that E12 has been reported in meningitis cases, but never in CNS cases?
Please indicate in the introduction some background on ECHO viruses and CNS disease as well as background on diganosis of EV infections with POC or real time assays or sanger or NGS. als indicate the prefered time form sample to results and if this algoritm fall within the acceptable time frame
in discussion: line 135: indicate cost per nucloetide for both sanger and NGS nowadays.
Author Response
Reviewer comments and Suggestions for Authors.
The manuscript has improved, but there are still a few issues that need to be adressed.
Please remove the figure based on the 4k tree. As indicated before due to reombination, complete genome trees for EVs is difficult. Instead please add the VP1 tree shown in the supp and add a 3Dpol tree. Remove all non E-12 strains and only include one other ECHO strain (eg E1 or E30) as outliers.
Authors’ answer: As advised by the reviewer, the phylogenetic tree was generated only with VP1 sequences from the Echovirus 12 genome recovered from GenBank database “Track changes” (lines 96-99) and figure 2, Appendix 1.
Line 118-120- This sentence in not correct. meningitis is a CNS related disease, yet the authors state that E12 has been reported in meningitis cases, but never in CNS cases?
Authors’ answer: The reviewer is perfectly right and the sentence has been corrected accordingly (Lines 119-122).
Please indicate in the introduction some background on ECHO viruses and CNS disease as well as background on diganosis of EV infections with POC or real time assays or sanger or NGS. als indicate the prefered time form sample to results and if this algoritm fall within the acceptable time frame
Authors’ answer: Please see amendments in accordance with this reviewer’s comments: “Track changes” (lines 25-26).
in discussion: line 135: indicate cost per nucleotide for both sanger and NGS nowadays.
Authors’ answer: The authors are now providing such an estimate in their laboratory. We estimated that current cost per nucleotide is 0.3€ for Sanger sequencing and 0.03 € for NGS sequencing with Illumina technology. This cost per nucleotide cannot be directly extrapolated in other laboratories, due to highly variable cost components, including the commercial policy of suppliers: see lines 136-140: “Indeed we have estimated that current cost per nucleotide is 0.3 € for Sanger sequencing and 0.03 € for NGS sequencing with Illumina technology; such cost cannot be directly extrapolated to other laboratories due to highly variable cost components, among which the commercial policy of suppliers or the infrastructure of the laboratory”. “Track changes” (lines 135-141).
Round 3
Reviewer 3 Report
The authors have addressed the comments adequately.